# The Importance of Topological Defects in Photoexcited Phase Transitions Including Memory Applications

**Dragan Mihailovic** [1,2] [iD]

1    Jozef Stefan Institute, Jamova 39, 1000 Ljubljana, Slovenia; dragan.mihailovic@ijs.si
2    CENN Nanocenter, Jamove 39, 1000 Ljubljana, Slovenia

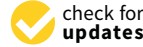

**Featured Application: Non-volatile information storage using metastable charge ordered states offers a low-energy ultrafast alternative to current memory technologies.**

**Abstract:** Photoinduced phase transitions have become a very important field of study with the advent of diverse time-resolved experimental techniques whose time resolution matches the electron, lattice, and spin relaxation dynamics associated with elementary excitations in quantum materials. Most techniques currently available rely on stroboscopic data-averaging over multiple transition outcomes. However, each time a transition takes place, fluctuations close to the time of the transition ensure that the phase transition outcome is different, with the emergence of different topological defect textures. In this paper, we briefly review the non-perturbative processes in selected charge-ordered quantum systems and the methods for their observation with different time-resolved techniques and scanning tunneling microscopy, which avoids the problem of averaging. The topological defect dynamics are seen to play an essential role in stabilizing emergent states in non-equilibrium transitions, appearing on different timescales as well as determining the emergent properties of the system. The phenomena are fundamentally important for understanding the fabric of matter in the Universe, as well as for possible applications in non-volatile memory devices.

**Keywords:** Photoinduced phase transitions; topological defects; domain walls; Kibble-Zurek phenomenon; time-resolved optical; time-resolved x-rays; photoexcited scanning tunneling microscopy; non-volatile memory

## 1. Introduction

Recent progress in time-domain techniques has made it possible to study the dynamics of elementary excitations under different non-equilibrium conditions and whence also the appearance of emergent order in real time [1–5] With low laser excitation intensities, the system is perturbed only slightly out of equilibrium, and the response is comparable to the information that one can obtain from frequency domain spectroscopies (the perturbative, or weak excitation regime). Different excitations can be directly distinguished by their respective lifetimes, so with different probes (including optical polarization analysis, X-ray or electron diffraction) very detailed information can be obtained on the system dynamics near equilibrium in many different materials, strongly correlated superconductors being perhaps the most researched. Beyond the low-excitation intensity regime, the possibility of driving the system through a phase transition (in the non-perturbative regime) brings the exciting prospect of monitoring the changing dynamics of the elementary excitations and thus the emergence of long-range order through the transition [6–9]. The early studies using multiple-pulse optical techniques have already revealed that the response in the emergent state is complicated by the

appearance of inhomogeneities—particularly topological defects [7]—that are quite difficult to study with conventional time resolved techniques that calculate an average over multiple transition outcomes.

In this paper we will discuss the importance of such topological defects in rapid phase transitions, both in the context of the Kibble–Zurek phenomenon in second-order symmetry breaking transitions [10–12], and in charge-driven topological transitions. In short-laser-pulse induced phase transitions, the time-evolution of the system is so rapid that different areas of the sample emerge from fluctuations through the phase transition with uncorrelated phase $\phi$ of the complex order parameter $\psi = Ae^{i\phi}$. By necessity, therefore, domains and associated topological defects must form between these areas, as noted originally for 2nd order transitions by Kibble [10–12]. However, the principle is quite general and extends to topological transitions as well, although the details are still not well understood.

We recently found that the emerging domain structure in such transitions is not necessarily random. In certain systems, the domains emerging after a non-equilibrium topological transition become long-range ordered [13]. Rapid phase transition experiments thus reveal quite rich complexity that requires new experimental methods beyond stroboscopic averaging in order to reveal the intricate details of the new emergent states. In this paper, we will discuss a few examples illustrating the physics on the basis of recent experiments on charge density wave and charge-ordered systems through 2nd order symmetry-breaking in rare-earth tritellurides and topological transitions in 1T-TaS$_2$, respectively.

Finally, the field of topological states emerging through non-equilibrium transitions has gained wider importance because ultrafast switching between different electronically ordered states can be used to store information, leading to ultra-low energy non-volatile ultrafast memory devices for which there is currently a great need. Comparison between charge density wave switching devices with existing ones, including phase-change memories and various magnetic storage devices, shows superiority in multiple areas that leads to realistic prospects for commercialization. Obviously, the search using ultrafast techniques for better and more convenient materials and systems is imperative, stimulating further research in the field.

## 2. Prototypical Charge-Ordering Materials and Methods: 3-Pulse and STM Techniques

The present paper discusses primarily layered chalcogenides that exhibit some form of charge ordering. Their main advantage is that they are amenable for study in the sense that both single particle and collective excitations are clearly visible. The transition metal dichalcogenides (TC$_2$), which we discuss, are typically variations on a theme based on a triangular 2D lattice, the detailed structure depending on the polytype (most commonly 1T and 2H). Rare-earth trichalcogenides (RX$_3$) have a layered structure with a square in-plane lattice [14], with similar dynamics. The materials exhibit very diverse charge orders, antiferromagnetic spin order (e.g., RX$_3$), or even quantum spin liquid properties [15]. In these experiments we concentrate on the topological defects that emerge upon non-equilibrium charge ordering. The prototype material with the most versatile properties discussed here is 1T-TaS$_2$, which undergoes a metal-to-incommensurate (IC) state transition followed by a discommensuration transition to a nearly commensurate (NC) state at 350 K, and eventually to a fully commensurate state below ~180 K. It also exhibits a new photoexcited hidden (H) state that was recently revealed by STM to have an unusual dual chiral order.

The methods used in the presented experiments are presented elsewhere [7,13]. Here, we only present an outline to help understand the presented phenomenon. The 3-pulse technique (Figure 1a) uses an intense laser pulse (the D pulse) to non-perturbatively destroy the equilibrium order, while a pump (P) and probe (p) pulses are used to measure the resulting time-evolution of the system. For a full understanding of the response, it is necessary to perform detailed modeling [7,16,17], but the main features of the technique are that it allows us to measure the single particle and coherent collective amplitude mode (AM) evolution through the transition. The inherent inhomogeneity of the excitation, due to the finite penetration depth and lateral extent of the beam, gives rise to domains, as shown schematically in Figure 1a. Since the lateral dimensions of the beam (100 um) are large compared with the depth (20–100 nm), the main effect is the formation of domains parallel to the surface (see Figure 1a).

These domains annihilate coherently, which is observed in the reflectivity dynamics of the collective mode, as shown in the next section. The technique is inherently stroboscopic, so the response is a temporal average over multiple transition outcomes initiated by each D pulse.

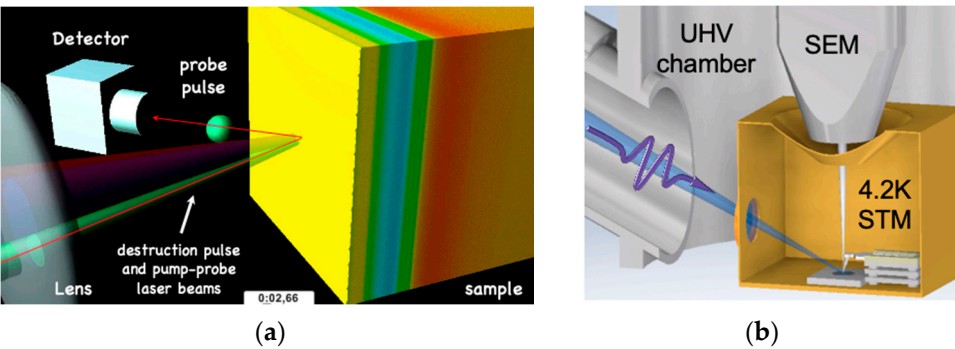

**Figure 1.** (**a**) A schematic picture of the domain structure that forms as a result of inhomogeneous excitation by the laser, as investigated by the 3-pulse technique. (**b**) A schematic of single-shot experiments within an STM chamber. An SEM microscope is used to align the tips and the beams on the sample.

For the study of domain wall dynamics, time-resolved X-ray diffraction [18,19]. and time-resolved electron diffraction [20–22] have been introduced. Although they do not directly show the topological defect structure in real space, the modelling gives evidence for the global behavior, including the relaxation dynamics.

With the use of single-shot techniques we can examine single transition outcomes in more detail. For this, it is important that the measurement time is shorter than the state lifetime, so the emergent state lifetime needs to be long, and preferably tunable with temperature. The only known such system so far is 1T-TaS$_2$, whose lifetime ranges from >$10^4$ years at low temperatures, to ~1 ms at 200 K [23]. Consequently, at low temperatures, very precise STM measurements can be performed to investigate the charge order, and particularly the domain structure in fine detail. A schematic of such an experiment is shown in Figure 1b.

## 3. Observation of Topological Defects on Different Timescales

We will discuss three different types of manifestations of topological defects that commonly arise in non-equilibrium phase transitions on different timescales: Coherent domain wall annihilation effects on a picosecond timescale that arise from inhomogeneous excitation, incoherent processes on intermediate timescales extending to tens of ps, and finally metastable defects stabilized by topological protection.

### 3.1. Coherent Domain Wall Annihilation in 3-Pulse Optical Experiments

The coherent reflectivity oscillations of the AM are a direct observer of the order parameter through a charge-density wave transition. The AM frequency shows a softening associated with the transition in 3-pulse experiments. The frequency also shows anomalies that can be understood by modelling the time-evolution of the order parameter $\psi$, through the transition with time-dependent Ginzburg–Landau theory [7]. Figure 2a shows the calculated response of the AM for the case of DyTe$_3$ as a function of time through the transition. The calculation takes into account the exponential profile of the laser photoexcitation due to the finite penetration depth of light into the sample. As a result of this inhomogeneity, domains form parallel to the surface which are clearly observable in the modelling. Here it is important to note that the model parameters are all experimentally measured, so the model predictions contain no adjustable parameters. The time-evolution of the AM spectra is shown in Figure 2b for the case of DyTe$_3$. After a short period of time we observed a signature of the soft mode emerging from zero, and hardening in frequency with time. After a long period of time

the frequency recovers, provided the system can relax in between subsequent pulse sequences. We also observed anomalous softening of the AM from the portion of the sample where $\psi$ was not fully suppressed by the D pulse (presumably due to inhomogeneity), but the details of this behavior are not fully understood within the simple model [7].

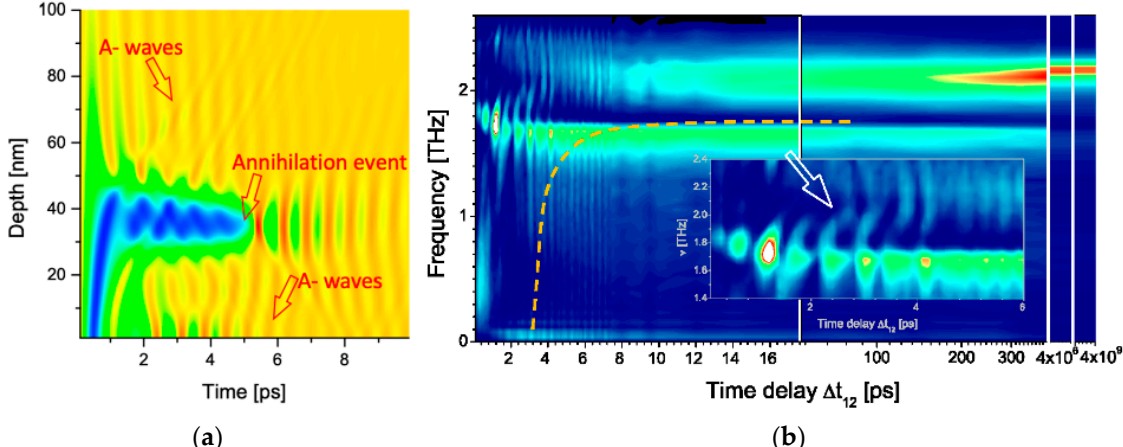

(**a**)                                                                                    (**b**)

**Figure 2.** Coherent generation and annihilation of domains in photoexcitation experiments. (**a**) The calculated order parameter (in color depicts the AM power spectrum) shows domains parallel to the surface as a result of inhomogeneous pumping. The domain wall annihilation is visible at t = 4.5 ps, resulting in the generation of A-waves. The A-wave that travels towards the surface can be detected in the optical response of the A mode frequency. (**b**) The power spectra of the oscillatory response for the collective mode in DyTe$_3$ as a function of time after the phase transition at $t = 0$. The soft-mode hardening is indicated schematically by the dashed line. The insert shows the anomalous behavior of the frequency as the A-waves emitted after domain wall annihilation reach the surface. (The arrow points to the anomalies).

The formation and subsequent annihilation of domains release amplitude waves, which cause anomalous line-shapes to appear in the collective mode spectra at delays of 2–5 ps with respect to the transition time. These correspond to the time of arrival of AM disturbances at the surface of the sample, which are predicted to occur upon annihilation of domain wall events [7]. These A-waves are unusual, soliton-like CDW amplitude excitations that travel with the velocity of sound away from the annihilation event, as shown in Figure 2b. They are detected at the surface by the anomalies that they cause in the frequency of the collective amplitude mode. By varying the parameters in the calculation, such as the penetration depth, one can convince oneself that the anomalous spectral features are indeed due to the A-waves. We note that the A-waves are finite frequency propagating modes that have no analogy in equilibrium systems. The observed effects are coherent, because the domains appear deterministically each time a D pulse is applied and only depend on the intensity of the pulse (i.e., how far the excitation penetrates into the sample). Increasing the fluence of the D pulse increases the number of domains that appear. Their annihilation is intrinsic, in the sense that the time-evolution of the system is determined by the TDGL dynamics, and does not seem to depend on external factors in a clean sample without extrinsic imperfections.

### 3.2. Inhomogeneous Defect Dynamics: Intrinsic and Extrinsic

The coherent response of the AM can be used to investigate the incoherent defect dynamics. Mertelj et al. [24] showed that the linewidth of the AM can be used to determine the dynamics of topological defects that relax incoherently. The main problem with such experiments is that the heating caused by the D pulse also causes broadening and a frequency shift on the AM. A significant effort was necessary to eliminate heating artefacts from the analysis as described in [24].

Two timescales were observed in the relaxation of topological defects created during a rapid quench of a charge-density-wave system through the electronic ordering transition. The defects were attributed to the Kibble–Zurek mechanism [24]. Using a three-pulse femtosecond optical spectroscopy technique and careful consideration of thermal processes, intrinsic topological defect annihilation processes were identified on a time scale of ~30 ps, and a possible signature of extrinsic defect-dominated relaxation dynamics was found at longer time scales. The non-thermal linewidth for the case of relaxation in two different samples of TbTe$_3$ is shown in Figure 3. The modulation of the AM due to the defects that annihilate incoherently is not detected directly by the stroboscopic technique. However, the incoherent topological defect dynamics give rise to a spatial inhomogeneity of the order parameter and a decoherence of the AM oscillations leading to an increased linewidth $\gamma_{AM}$ for $t < 30$ ps. Concurrently, the defects give rise to a softening of the collective mode frequency $\omega_{AM}$ because of the suppression of the order parameter that they cause [24]. The increase of the effective damping at shorter times, as shown in Figure 3, is attributed to the inhomogeneity of the order parameter. As the experiments are stroboscopic, no detailed information beyond the relevant timescales is available, and no information on the defect structures could be deduced.

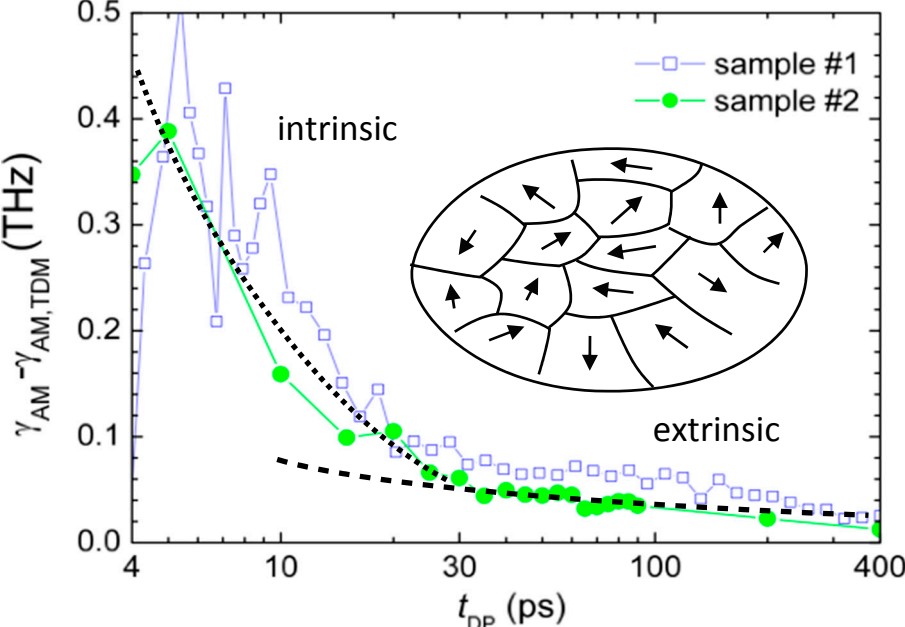

**Figure 3.** Incoherent relaxation of domains in photoexcitation experiments. The time-dependence of the excess linewidth $\gamma_{AM} - \gamma_{AM}^{TDM}$ due to the annihilation of topological defects. Here, $\gamma_{AM}$ is the full linewidth, and $\gamma_{AM}^{TDM}$ is the thermal linewidth. Data for two different samples show a difference primarily in the long-time behavior beyond 30 ps, which appears as an offset. The insert depicts the laser photoexcitation spot where the phase of the order parameter is represented schematically by arrows.

### 3.3. Topological Defect Dynamics Observed by X-rays and Low-Energy Electron Diffraction

In principle, diffraction experiments can give information on the detailed structure only if the phase can be recorded. Although this is not yet possible with current state-of-the-art techniques, the time-evolution of the diffraction intensity is more informative than the optical experiments discussed in the previous section, particularly when combined with modeling. Laulhe et al. have shown the growth of domains in the nearly commensurate phase of 1T-TaS$_2$ to follow Lifshitz–Allen–Kahn dynamics on a timescale of ~100 ps [18]. The observed CDW correlation length in the incommensurate state was shown to be described well by the expected power law $\xi_C \propto t^{1/2}$. Lantz et al. [19] also observed a marked anisotropy the size of the photoinduced domains of the I-phase of 1T-TaS$_2$, which appeared to grow self-similarly, such that their shape remained unchanged throughout the growth process.

Recent ultrafast low-energy electron diffraction (ULEED) experiments [20] show coarsening that follows a power-law scaling of the correlation length, driven by the annihilation of dislocation-type topological defects of the charge-ordered lattice. The technique, in combination with modelling on the level of time-dependent Ginzburg–Landau theory, allows detailed interpretation of the observed electron diffraction patterns, and in particular, interpretation of the dynamics of the domain formation. Zong et al. [22] independently tracked the amplitude and phase dynamics of the CDW by a combination of techniques, finding ~1 ps recovery of the CDW amplitude, followed by a slower re-establishment of phase coherence dictated, which they attributed to the presence of topological defects. The timescales of these experiments on the NC–IC transition in 1T-TaS$_2$ are in good agreement with optical experiments on the IC transition in TbTe$_3$ discussed in the previous section.

As much as these techniques agree on the dynamics, they are all limited by spatial and temporal stroboscopic averaging. Features in the structure that are non-periodic, such as dislocations, or that appear on multiple length scales, are not discernible in the diffraction experiments.

### 3.4. Topological Defects Observed in Single-Shot Experiments by STM

In 2D systems in which the final state lifetime is long, such as 1T-TaS$_2$ [23], we can investigate the spatial non-periodic structures with STM, obtain electronic density of states maps, and observe quasiparticle interferences that can be used to reconstruct the shape of the Fermi surface [25].

Here, we will focus on the metastable structures observed after photoexcitation of the hidden phase in 1T-TaS$_2$. Remarkably intricate patterns emerged after a single shot: The uniform charge order is disrupted, and charged domains form, which accommodate the photoinduced charge (Figure 4) [26].

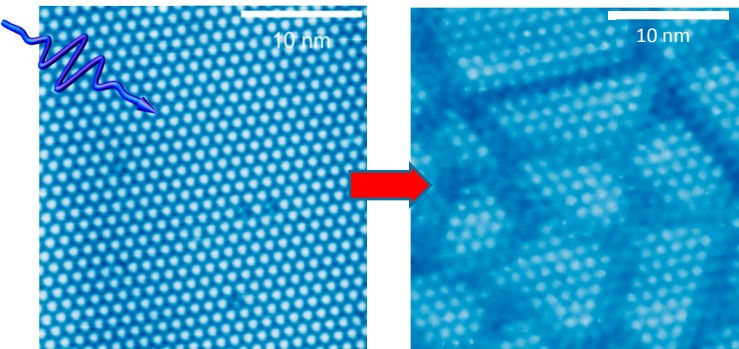

**Figure 4.** The formation of domains in the photoexcited hidden state (right) after excitation of the commensurate charge ordered state (left) in 1T-TaS$_2$ with 1mJ/cm2, 30 fs pulses at 800 nm [26].

While this state looks remarkably similar to the nearly commensurate state or the state created by direct charge injection through an STM tip [28,29], detailed analysis by STM shows that they have subtle differences which cannot be easily discerned by other techniques [13]. In particular, the charge order in adjacent domains is displaced in such a way that it forms vortices (Figure 5). Defining the displacement vector field $\vec{\mathcal{D}}$ (Figure 5), we see that the angle of the charge displacement vector between domains, $\alpha$, shows a chiral pattern in which the vortexes are arranged in a periodic lattice with a wavelength ~70 nm. This structure is unique to the photoinduced hidden state—the nearly commensurate equilibrium state which exists at room temperature shows a different periodicity, while STM-switched domains do not display any long range order [27]. We attribute the appearance of the mesoscopic order on the ~70 nm length scale to the relatively homogeneous excitation by laser over the laser spot (~200 μm), which is not the case with STM tip excitation.

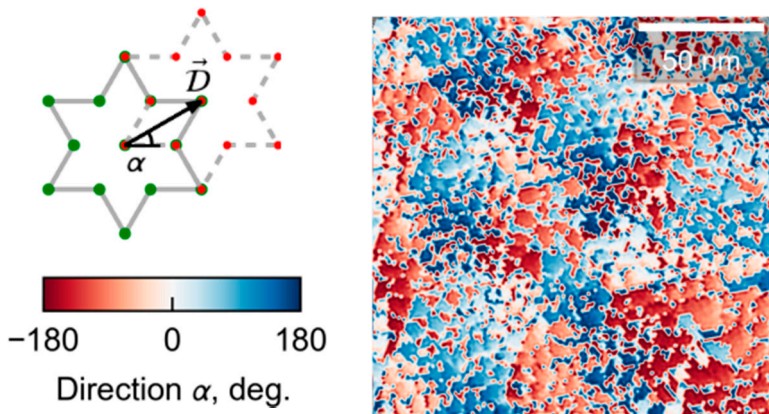

**Figure 5.** The vortex structure of the charge ordered domains in the photoexcited hidden state of 1T-TaS$_2$. The legend shows the definition of the displacement vector $\vec{\mathcal{D}}$ and the angle of relative displacements between domains $\alpha$. The displacements show a distinct long-range ordered vortex lattice with a period of $\sim$70 nm.

The detailed mechanism for switching dynamics induced by ultrashort light pulses is related to the relaxation path of the photoexcited electrons and holes created by the absorption of photons. Initially, this does not result in photodoping: The system remains charge-neutral. However, if the energy relaxation dynamics of electrons and holes is different, then there is a time window—typically in the order of a few ps—within which one type of carrier has already thermalized, while the other has not, resulting in transient photodoping. During this time window, there may be an opportunity for the carriers to form a domain structure, where domain walls accommodate the excess transient charge. However, to form the periodic chiral charge patterns observed in the hidden state of 1T-TaS$_2$ [27] (Figure 5), additional ordering of domains is likely on intermediate timescales.

## 4. The Application of Emergent Metastable States to Non-Volatile Cryo-Memory Devices

The photo doping hypothesis, by which the cause of metastability are the additional carriers, can be confirmed by performing carrier injection through an electrical contact [15,23] or an STM tip [28,29]. Such experiments not only confirm that carrier injection causes domain wall formation [13], but also indicate that the accompanying change in resistivity makes it possible to construct a novel kind of non-volatile all-electronic ultrafast cryo-memory device based on the phenomenon. The metastability which is necessary for non-volatile device operation, is provided by the topologically protected defects, such as dislocations, that are created in the charged-induced quench. Switching resistance in device configuration was demonstrated using 40 ps electrical pulses, which is unmatched by any other technology at the time of writing, with a measured switching energy of <200 fJ/bit [30] using a relatively large (2 μm) device. Reducing the device dimensions and using shorter pulses towards the 0.5 ps limit [31] would make such memories compatible with superconducting flux quantum devices, reducing the switching energy well below 1 fJ/bit, potentially opening the way to a viable non-volatile cryo-memory technology that would replace current energy-inefficient and relatively slow CMOS memory in cryocomputing applications [32]. The other potential advantages of such a device based on 1T-TaS$_2$ are their relative simplicity, low-temperature operation, and endurance. The latter is a consequence of the fact that only electrons are moved around into different configurations. Since no atoms are moved beyond their CDW positions, we do not expect the endurance problems that occur in the filamentary paths of more conventional memristors.

## 5. Discussion

While it is clear that the functional properties of phases emerging after non-equilibrium transitions may be strongly dependent on the structure of topological defects, controlling the different kinds of

defect order is a significant challenge. For example, in coherent domain wall creation, the fluence of photoexcitation can be used to control how many domain layers are created parallel with the surface, with concentric circles arising from lateral intensity profile of the laser spot. On the other hand, the chiral structures that appear in the hidden state of 1T-TaS$_2$ are difficult to control at present. Our present understanding is that they arise as a result of the particular shape of the non-equilibrium Fermi surface, which forces a mesoscopic order [13], but the details still need to be investigated further. However, the mirror domains walls that appear upon breaking of symmetry associated with the NC transition in 1T-TaS$_2$ were found to be controllable to some extent, with probabilities for locally injecting and removing them tunable by pulse energy and temperature [21].

The number of topological defects created by photoexcitation appears to be related in some way to the laser fluence, which is in turn related to the number of photoexcited carriers that localize in the domain walls. This would imply that we might be able to control the density of domain walls by light fluence. However, the experiments in 1T-TaS$_2$ indicate that there is a well-defined fluence threshold for switching from the C to the H state. The implication is that critical behavior governs the transition between the different states, rather than a continuous change of domain wall density. Since the ordering proceeds with the system at a finite (non-equilibrium) temperature, there is no quantum critical point transition.

The charge transport in the domain textured states is currently still an open problem. The conventional picture of domain walls acting as "rivers of charge" is not confirmed by experiments, which shows that both domains and domain walls become conductive, and shows gapless STS spectra [28,29]. The domain size is larger than the proximity length (1–2 nm), so we cannot attribute the gapless behavior within the domains to a proximity effect of the metallic domain walls. The electronic structure of such textured states, and particularly the transport through such textures, is still not understood at the present time and represents an open and interesting problem. In particular, the apparent coexistence of itinerant metallic transport and localized charges observed by STM is not new here, and has been a paradigm in superconducting cuprates since the mid 1990s [33].

Finally, it is worth noting that topological defects, which stabilize the "hidden" state, may be useful for stabilizing photoinduced superconductivity, and particularly, in extending the lifetime of such states to the extent that superconductivity may actually be proven with measurements of four-probe resistance and diamagnetism. Our recent studies of doped superconducting dichalcogenides show domain wall coexistence with superconductivity in 1T-TaSeS, so the appearance of a metastable superconducting state in photoinduced chalcogenides may not come as a complete surprise.

## 6. Conclusions

The study of the microscopic nature of the emergent textures is clearly of wider fundamental interest. For single-outcome phase transitions, single shot experiments appear to be the essential research tool, and photoexcited STM is currently the most versatile. Unfortunately, time-resolved STM on femtosecond timescales is unlikely to be realizable in single shot mode, so systems with extended and tunable lifetimes remain the primary option, and 1T-TaS$_2$ is the first such material available.

We may expect the first potential use of metastable topologically protected states in cryogenic memories in the near future, subject to technological compatibility and other practical considerations, but the search for new metastable states emerging through ultrafast transitions is still at an early stage, particularly with regard to detailed understanding of the underlying phenomena. The search for new systems displaying metastability is deemed essential for progress in this area.

Ideally, non-volatile memory devices should have a very small switching energy, high speeds, and increasingly, low-temperature operation capability. The most important phenomenon for potential non-volatile memory devices is metastability, where the prevailing mechanism outlined in this paper is related to the formation of textures and associated diverse topological defects emerging through a phase transition. The accompanying large change resistance, such as that observed in 1T-TaS$_2$, makes the material ideal for low-temperature non-volatile memory applications. Unipolar write and erase

cycles [30], THz switching speed [31], and an all-electronic mechanism that ensures low programming energy, may appear in other systems provided that the domain wall structure is metastable. Indeed, the doping mechanism that gives rise to domain wall formation may be considered quite generic and not limited to a specific material, provided that the band structure is sufficiently asymmetric to result in a charge-asymmetric relaxation, so that temporary photodoping by the carriers that populate states at the Fermi level may localize in domain walls. The choice of suitable material may, thus, be related to the band structure features in the vicinity of the Fermi energy and significant band-asymmetry. In other words, relaxation of either electrons or holes should ideally be hindered by the presence of a gap in the density of states on either side of the Fermi level. Based on this idea, we may be able to search for new materials which exhibit metastable textures and non-volatile memory device functionality.

## 7. Patents

1. Vaskivskyi, I.; Mihailovic, D.D.; Mihailovic, I.A. Switchable Macroscopic Quantum State Devices and Methods for Their Operation. U.S. patent 9818479B2 (14. Nov. 2017).
2. Stojcevska, L.; Mertelj, T.; Vaskivskyi, I.; Mihailović, D. Ultrafast Quench Based Nonvolatile Bistable Device. U.S. patent 9,589,631B2 (7 Mar 2017).

**Funding:** This research was funded by ERC AdG "TRAJECTORY", GA 329 602 and the Slovenian research agency ARRS.

**Acknowledgments:** The author would like to acknowledge I. Vaskivskyi, Y. Gerasimenko, T. Mertelj, I. Ravnik, P. Karpov, S. Brazovskii and V.V. Kabanov for contributing to various published works reviewed in this paper.

**Conflicts of Interest:** The author declares no conflict of interest.

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
