# Peer review of "The Importance of Topological Defects in Photoexcited Phase Transitions Including Memory Applications"

_applsci, doi:10.3390/app9050890_

Round 1

Reviewer 1 Report

The author provides an overview of recent experimental technique about time-resolved measurement. The new method avoids the problem of averaging over multiple transition outcomes, thus it is useful for the systems in which the domains emerging after a non-equilibrium topological transition become long-range ordered. The author mainly reviewed on the Charge Density Wave and charge ordered systems through 2nd-order symmetry-breaking transition in rare-earth tritellurides and topological transitions in 1T-TaS2. I am happy to recommand the manuscript to be accepted for publication after a minor revision: There are a lot of errors/typos in the references. Some of the references are missing journal title or the article numbers, e.g.[6], [13], [19], [27], etc.

Author Response

We wish to thank the reviewer for his/her kind remarks and sincerely apologise for the errors in the automated reference list, which we have corrected in the revised manuscript.

Reviewer 2 Report

The paper presents important results and should be accepted 1) after some minor editing and 2) more conclusions.

1) Rather than “I will discuss”, best to use present tense and also not refer to “I”, e.g. “it is discussed” rather “I will discuss”. On the other hand expressions like “The present paper discusses” are appropriate.

2) The author needs to expand the "Conclusions" section, it is really too short, the author should explain more the significance and future applications:

"We may expect the first potential use of metastable topologically protected states in cryogenic memories in the near future, subject to technological compatibility and other practical considerations. But the search for new metastable states emerging through ultrafast transitions is still at an early stage, particularly with regard to detailed understanding of the underlying phenomena. The search for new systems displaying metastability is deemed essential for progress in this area."

Author Response

We thank the reviewer for his kind remarks and suggestions. 

Ad 1) We have made changes to conform to the suggested style. 

Ad 2) We have expanded on the Conclusions as suggested. This is a welcome opportunity to discuss possible applications and we are grateful to the reviewer for suggesting this explicitly. An additional paragraph was added to explain the significance and future applications.